# CUC/auxin patterning of decanalised petal number in *Cardamine hirsuta*

Léa Rambaud-Lavigne[1,2] , Marie Monniaux[1,3] , Zi-Liang Hu[1,4] ,
Sarah McKim[5,6] and Angela Hay[1]

[1]Max Planck Institute for Plant Breeding Research, Köln, Germany; [2]Laboratory for Plant Reproduction and Development, CNRS, INRAE, ENS Lyon, University of Lyon, Lyon, France; [3]Evo-Eco-Paleo, CNRS, University of Lille, Lille, France; [4]Center for Plant Systems Biology, Ghent, Belgium; [5]Plant Sciences Department, University of Oxford, Oxford, UK; [6]Division of Plant Sciences, School of Life Sciences, University of Dundee at the James Hutton Institute, UK

## Original Research Article

CUC/auxin patterning of decanalised petal
number in *Cardamine hirsuta*. *Quantitative
Plant Biology*, **6:**e29, 1–14

**Keywords:**
Cardamine hirsuta; CUC genes; flower
development; Petal number.

**Corresponding author:**
Angela Hay;
Email: hay@mpipz.mpg.de

**Associate Editor:**
Dr. Daphné Autran

## Abstract

Petal number is highly canalised in the four-petalled flowers of Arabidopsis. This trait is decanalised in the closely related species *Cardamine hirsuta*, such that petal number varies from zero to four between individual flowers and in response to natural genetic and environmental variation. Loss of robustness was traced to divergence of the MADS-box transcription factor *APETALA1* in *C. hirsuta*, resulting in loss of epistasis over alleles that cause petal number to vary. How petal formation is patterned in these decanalised flowers is an open question. Here we use genetics and quantitative imaging to investigate how a key patterning module, comprising CUP-SHAPED COTYLEDON1,2 (CUC) transcription factors and auxin, regulates petal formation in *C. hirsuta*. We show that auxin activity maxima are positioned in inter-sepal boundaries, rather than on the floral meristem, rendering petal initiation sensitive to the space available between sepals, such that growth variation influences petal number variation.

## 1. Introduction

Development is the process through which reproducible forms unfold from a genetic blueprint. This reproducibility in the face of natural genetic and environmental variation is often explained as a buffering of the developmental process. Waddington used the concept of canalisation to explain the general observation that wild-type forms are better buffered to variation than mutant forms (Waddington, 1942). Underpinning this concept is the control of developmental programmes by networks of genes connected by regulatory interactions. Canalisation is an emergent property of such complex gene regulatory networks (Siegal & Bergman, 2002).

The reproducible form of Arabidopsis (*Arabidopsis thaliana*) flowers is controlled by the combinatorial action of MADS-box transcription factors acting in specific whorled domains in the floral meristem (Coen & Meyerowitz, 1991). According to the floral quartet model, petals form on the floral meristem via the activity of a tetrameric protein complex containing APETALA1 (AP1), APETALA3, PISTILLATA and SEPALLATA MADS-box transcription factors (Honma & Goto, 2001). The combination of these transcription factors activates a petal-specific gene regulatory network that is itself embedded in a larger network controlling flower formation (Chen et al., 2018). Regulatory interactions within this gene network ensure that four petals form reproducibly in Arabidopsis flowers in spite of natural genetic variation (Monniaux et al., 2018). Therefore, petal number is robust in Arabidopsis.

By contrast, petal number is decanalised in a close relative of Arabidopsis called *Cardamine hirsuta*. Individual flowers on a single plant display a variable number of petals between zero and four, and this phenotype varies in response to natural genetic and environmental variation (McKim et al., 2017; Monniaux et al., 2016; Pieper et al., 2016). Therefore, petal number is not robust in *C. hirsuta*. This loss of robustness was traced to divergence of the MADS transcription factor *AP1*, which relaxed its epistasis over quantitative trait loci (QTL) that cause petal number to vary (Monniaux et al., 2018). Previous work showed that transferring the Arabidopsis *AP1*

genomic locus into *C. hirsuta* was sufficient to canalise petal number by masking the phenotypic effects of at least nine QTL in the *C. hirsuta* genome (Monniaux et al., 2018). Therefore, *AP1* divergence in *C. hirsuta* altered its epistatic interactions with genes in the floral regulatory network and was the likely mechanism by which cryptic variation was exposed in *C. hirsuta*, contributing to the evolution of variable petal number.

Epistasis plays a large role not only in trait evolution, but also in domestication and breeding. Genomes contain vast stores of potential cryptic variants, whose phenotypic effects are only exposed after mutating genes in epistatic interactions with them (Bergman & Siegal, 2003; Paaby & Rockman, 2014). The exposure and subsequent selection for or against cryptic variants have helped shape developmental traits during crop domestication and modern breeding (Soyk et al., 2020). For example, in tomato, phenotypic effects of cryptic alleles of MADS-box genes, with both negative and positive effects on yield, were exposed by epistatic interactions during breeding for jointless fruit (Soyk et al., 2017, 2019). Therefore, complex gene regulatory networks buffer developmental processes and modification of these regulatory interactions, through evolution, domestication or breeding, contributes to phenotypic variation.

Arabidopsis flowers achieve a reproducible number of floral organs through the precise control of gene expression domains in space and time (Neumann et al., 2022; Refahi et al., 2021). How such patterning processes function in decanalised *C. hirsuta* flowers to produce a variable number of petals is unclear. To gain insight into this process, we studied the activity of a patterning module comprising the growth-promoting hormone auxin and growth-repressing transcription factors CUP-SHAPED COTYLEDON1,2 (CUC1,2) that belong to the NAM-ATAF-CUC (NAC)-domain family (Aida et al., 1997). This patterning module is broadly important for plant development, and functions in various developmental contexts and different species (Berger et al., 2009; Bilsborough et al., 2011; Furutani et al., 2004; Galbiati et al., 2013; Heisler et al., 2005; Kierzkowski et al., 2019). In Arabidopsis flowers, CUC1,2 transcription factors establish boundaries that demarcate the positions of four petal primordia on the adjacent floral meristem (Baker et al., 2005), which are marked by the formation of auxin maxima (Chandler et al., 2011; Lampugnani et al., 2012; 2013; Monniaux et al., 2018). How CUC transcription factors instruct the patterning of auxin maxima has been previously described in the context of leaf development (Bilsborough et al., 2011; Hu et al., 2024; Kierzkowski et al., 2019). In *C. hirsuta*, CUC1 regulates the expression of auxin-related genes, including the direct transcriptional activation of WAG serine/threonine AGCVIII kinases that regulate the polarity of the plasma membrane-localised PIN-FORMED1 (PIN1) auxin transporter (Hu et al., 2024). Polar auxin transport relies on the export of auxin out of cells by PIN transporters to create auxin asymmetries that are central to plant growth and development. Specifically, the convergence of PIN1 polarities create a maximum auxin response that can be read out as activation of DR5 or degradation of DII reporters (Brunoud et al., 2012; Ulmasov et al., 1997) and trigger outgrowth. Thus, through the direct regulation of auxin targets, the CUC1 transcription factor in *C. hirsuta* contributes to PIN1 repolarisation, enabling new PIN1 convergences that underlie organogenesis (Hu et al., 2024).

In this study, we used genetics, time-lapse confocal imaging and quantitative analysis of early flower development (Rambaud-Lavigne & Hay, 2020) to investigate CUC/auxin patterning of decanalised petal number in *C. hirsuta*. We found that *CUC1,2* genes regulate petal number by forming boundary regions between

sepals and promoting the convergence of PIN1 polarities to create auxin maxima in these boundaries. The positioning of auxin activity maxima in the sepal whorl, rather than in the adjacent whorl on the floral meristem, is a key difference to Arabidopsis. Therefore, the same patterning module is deployed differently to initiate petals in *C. hirsuta* versus Arabidopsis. Modifying cellular growth, by manipulating CUC activity or ambient temperature, altered the amount of space between sepals in which discrete auxin maxima formed to trigger petal initiation. In this way, growth is integral to the patterning process controlling petal initiation in *C. hirsuta* and easily influenced by environmental and genetic variation, resulting in variable patterning outputs.

## 2. Results

### 2.1. CUC genes regulate petal number in C. hirsuta by modifying growth and auxin patterning

To investigate the role of *CUC* genes in regulating petal number in *C. hirsuta*, we analysed different genotypes with loss or gain of *CUC* function. In order to reduce CUC activity, we used *cuc2-1* mutants (Rast-Somssich et al., 2015) and the previously described *2x35S::MIR164B C. hirsuta* line where overexpression of the Arabidopsis *MICRORNA164B* gene results in post-transcriptional degradation of both *CUC2* and *CUC1* (Blein et al., 2008). To increase CUC activity, we transformed *C. hirsuta* with *5mCUC1*: a previously described miR164-resistant genomic clone of Arabidopsis *CUC1* (Mallory et al., 2004). Reduced CUC activity resulted in loss of petals in both *cuc2-1* and *2x35S::MIR164B* flowers (Figure 1a,b). Adjacent sepals were fused to varying extents in *2x35S::MIR164B* (Supplementary Figure S1), but not *cuc2-1* flowers. *Cardamine hirsuta 5mCUC1* flowers with increased CUC1 activity had significantly more petals than the wild type (Figure 1a,b). *5mCUC1* flowers had a median petal number of four and up to six petals were occasionally observed (Figure 1a,b). These phenotypes are consistent with previous observations in Arabidopsis (Laufs et al., 2004; Mallory et al., 2004), and indicate that the variable petal number in wild-type *C. hirsuta* can be modulated by altering the level of CUC activity.

To investigate the effects of CUC activity on petal number, we considered the dual role of *CUC* genes to both specify boundary regions and to promote organogenesis via the formation of auxin maxima. We observed that boundary regions between sepals were reduced in *2x35S::MIR164B* and enlarged in *5mCUC1* flowers, when compared to wild-type flowers (Figure 1c and Supplementary Figure S1). To quantify these phenotypes at a cellular level, we used time-lapse confocal imaging and MorphoGraphX software (Barbier de Reuille et al., 2015) to compute the amount of cellular growth that occurred during 10 h of development in stage 4 flowers (Figure 1c). In *2x35S::MIR164B* flowers, we found that cellular growth was increased in the sepal whorl, particularly in the regions between sepals, resulting in overgrowth of the inter-sepal boundaries (Figure 1c). Conversely, cellular growth was reduced in broad regions between sepals in *5mCUC1* flowers, creating larger inter-sepal boundaries and narrower sepals (Figure 1c). Thus, altering the level of CUC activity influences the amount of space available between sepals for petals to form in *C. hirsuta* flowers.

We have previously shown that peaks of auxin activity maxima associated with initiating petals are often absent from the floral meristem, but are rather located in the inter-sepal boundaries of *C. hirsuta* flowers (Monniaux et al., 2018). This is in stark contrast to Arabidopsis where these auxin activity peaks are always located on

the floral meristem (Figure 4b) (Chandler et al., 2011; Lampugnani et al., 2012; Lampugnani et al., 2013; Monniaux et al., 2018). To investigate whether the formation of auxin activity maxima varies in response to CUC activity in *C. hirsuta* flowers, we analysed a *DR5v2::nls:3xVenus* (DR5) reporter in our time-lapse series (Figure 1d). In *5mCUC1* flowers, we consistently observed discrete foci of DR5 expression in each of the four inter-sepal boundaries (Figure 1d). On the other hand, foci of DR5 expression were absent from inter-sepal boundaries of *2x35S::MIR164B* flowers (Figure 1d). To visualise DR5 distribution more quantitatively, we plotted DR5 signal intensity radially in the approximately circular floral primordia (Figure 1e–h). The most intense DR5 peaks are at sepal tips in all flowers, and petal initiation regions lie between these peaks (pink shading, Figure 1e–h). In wild-type flowers, weak DR5 signal of varying intensity was observed between sepals (Figure 1e), but these weak, variable peaks were barely evident when the signal was averaged across 10 different samples (Figure 1f). DR5 peaks were absent from the inter-sepal boundaries of *2x35S::MIR164B* flowers (Figure 1g), whereas strong peaks of DR5 signal were present between each sepal in *5mCUC1* flowers (Figure 1h). Therefore, gain or loss of CUC activity is associated with the presence or absence of auxin activity peaks in inter-sepal boundaries. In summary, *CUC1,2* genes regulate *C. hirsuta* petal number by specifying slow-growing boundaries between sepals that are competent to form auxin maxima.

## 2.2. Ambient temperature regulates petal number in C. hirsuta

We have previously shown that petal number in wild-type *C. hirsuta* is regulated by differential growth and maturation of floral buds in response to ambient temperature (McKim et al., 2017). Slower growth and prolonged maturation time at lower temperatures produced larger boundaries between sepals and more petals (McKim et al., 2017). Since increased CUC activity also created larger inter-sepal boundaries and more petals (Figure 1), we investigated the interaction between *CUC* genes and ambient temperature. When we grew *CUC* loss- and gain-of-function genotypes at 15°C, we found a significant increase in petal number in all genotypes, compared to plants grown at 20°C (Figure 2a). In particular, *cuc2-1* and *2x35S::MIR164B* flowers produced a median number of three petals at 15°C compared to zero petals at 20°C (Figure 2a). This suggests that the growth and maturation processes regulated by ambient temperature are not dependent on CUC activity. In line with this conclusion, significant differences in petal number between all *CUC* genotypes were observed at both 15°C and 20°C (Figure 2a).

We reasoned that the increase in wild-type petal number at 15°C might be associated with more regular formation of DR5 peaks between sepals, similar to our findings in *5mCUC1* flowers (Figure 1d,h). To investigate this, we used time-lapse confocal imaging to compare DR5 expression in stage 4 flowers of plants grown at 15°C or 20°C (Figure 2b). We observed more discrete foci with higher DR5 signal intensity in inter-sepal boundaries at 15°C (Figure 2b). We also measured the amount, duration and distribution of growth in these flowers (Figure 2c). The duration of growth between similar developmental stages was much longer at 15°C, indicating slower growth, and cells with low amounts of growth were distributed in broader regions between sepals at 15°C compared to 20°C (Figure 2c), matching previous results (McKim et al., 2017). Therefore, DR5 peaks form more consistently in the larger inter-sepal regions of flowers grown at 15°C, similar to *5mCUC1* flowers.

To determine the pattern of *CUC2* transcription in *C. hirsuta* flowers, and whether it is altered by ambient temperature, we analysed stage 4 flowers of a *C. hirsuta pChCUC2::3×GFP* transcriptional reporter (*ChCUC2*) grown at 20°C and 15°C (Figure 2d,e). At this stage, *ChCUC2* is expressed between sepals and excluded from sepals (Figure 2d). At 20°C, the intensity of *ChCUC2* expression varied between different inter-sepal regions (Figure 2e), and showed a broad, background level of expression throughout the floral meristem (Figure 2d). At 15°C, the signal intensity of *ChCUC2* was more similar between each sepal (Figure 2e), with maximum intensity at the boundary between the sepal whorl and the meristem (Figure 2d). *ChCUC2* expression was absent from the floral meristem at 15°C, creating a sharper differential between the expression in inter-sepal boundaries and lack of expression throughout the rest of the floral primordium (Figure 2d,e).

Altogether, these results indicate that DR5 and *ChCUC2* resolved to expression domains with less overlap in the larger inter-sepal boundaries formed in *C. hirsuta* flowers at 15°C compared to 20°C. Within each inter-sepal boundary, *ChCUC2* expression was focused adjacent to the floral meristem (Figure 2d) while DR5 maxima formed towards the periphery of the flower (Figure 2b). These expression patterns were associated with a significantly lower coefficient of variation for wild-type petal number at 15°C compared to 20°C (Figure 2h). Therefore, more space between sepals seems to allow more reproducible patterning of petal initiation.

To test whether increasing CUC activity specifically in inter-sepal boundaries can reduce the variability in *C. hirsuta* petal number, we generated *pPTL::5mCUC1* transgenics. We had previously shown that an Arabidopsis *PETALLOSS* (*PTL*) transgene expressed specifically in inter-sepal boundaries of *C. hirsuta* flowers (Monniaux et al., 2018). Therefore, we used this *PTL* promoter to ectopically express *5mCUC1* (Figure 2f). In plants grown at 20°C, petal number was increased in *pPTL::5mCUC1* flowers (Figure 2g and 3.4 ± 0.8 petals vs. 2.3 ± 1.1 petals in WT, mean ± SD) and the coefficient of variation was significantly lower (Figure 2h). Therefore, petal number variation was reduced by specifically increasing CUC1 activity in the inter-sepal domains of *C. hirsuta* flowers.

## 2.3. AUX1 and PIN1 localise to petal initiation regions in C. hirsuta flowers

To further investigate the patterning of petal initiation by auxin in *C. hirsuta* flowers, we analysed expression of the auxin efflux protein PIN1 and the auxin influx protein AUX1 (Bennett et al., 1996; Galweiler et al., 1998). Both proteins had been previously shown to play a role in the local accumulation of auxin that triggers petal initiation in Arabidopsis (Lampugnani et al., 2013). To localise PIN1, we used a previously described *C. hirsuta pChPIN1::ChPIN1:eGFP* (ChPIN1) fusion protein that complements *C. hirsuta pin1* mutants (Hu et al., 2024). To localise AUX1, we generated a functional *C. hirsuta pChAUX1::ChAUX1:YFP116* (ChAUX1) fusion protein with an in-frame fusion of YFP at amino acid 116 of *ChAUX1*, as described for Arabidopsis AUX1–YFP fusions (Swarup et al., 2004). We quantified cellular signal of ChAUX1 and ChPIN1 to compare expression at the tissue level in stage 4 flowers of wild-type plants grown either at 20°C or 15°C (Figure 3a–f). At this stage, ChAUX1 is expressed in inter-sepal regions, where petals initiate, as well as the tips of medial sepals and the floral meristem (Figure 3a). Ambient temperature had little effect on ChAUX1 expression (Figure 3a–c). In contrast, ChPIN1 expression was increased at 15°C with the most intense signal in inter-sepal regions (Figure 3d–f).

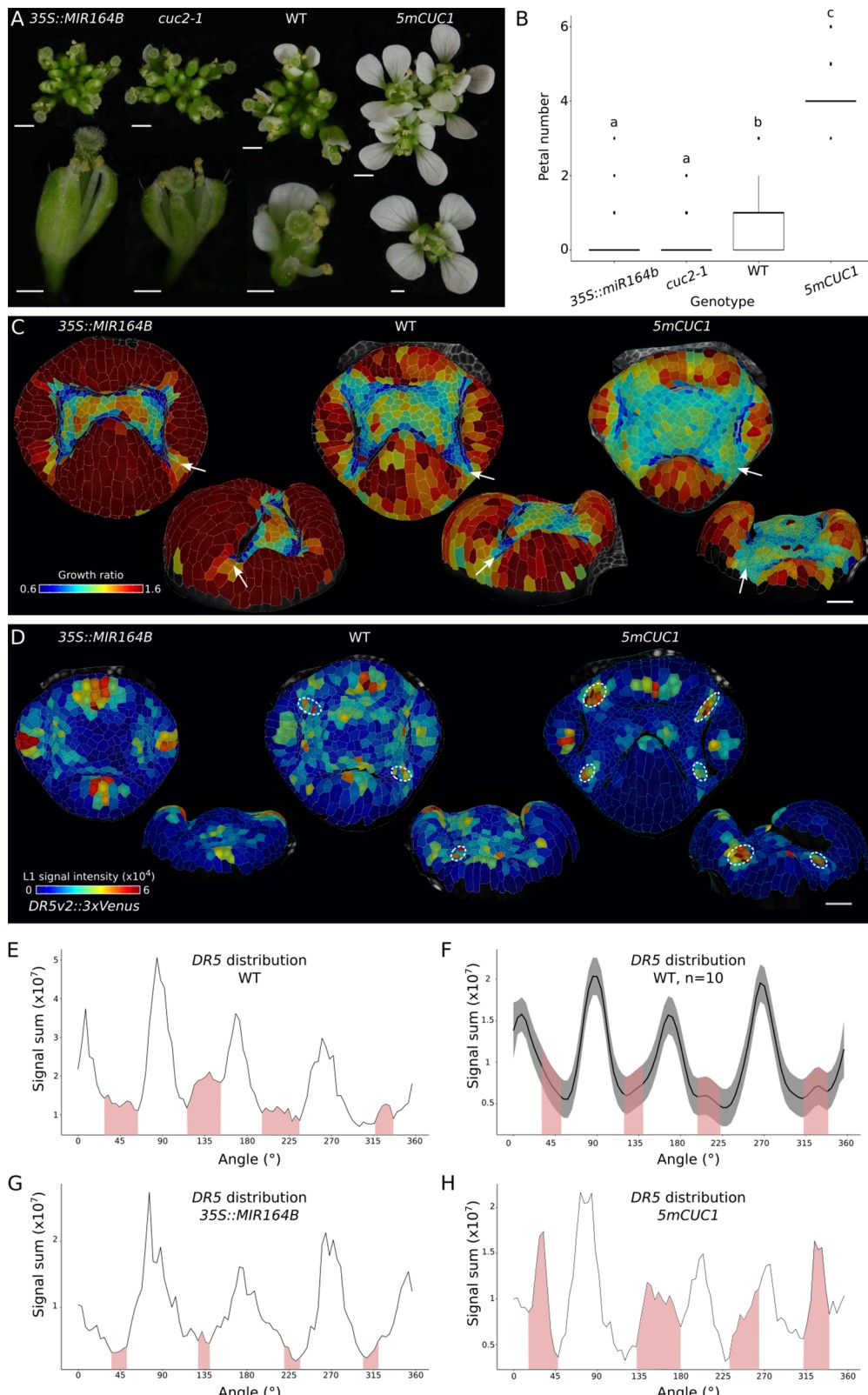

**Figure 1.** CUC activity regulates petal number in *Cardamine hirsuta* by modifying patterns of growth and auxin maxima. (a) Representative inflorescence (top) and flower (bottom) of *C. hirsuta 2x35S::miR164b*, *cuc2-1*, WT and *5mCUC1*. (b) Boxplots of average petal number in *C. hirsuta 2x35S::miR164b* (*n* = 84 flowers, four plants), *cuc2-1* (*n* = 127 flowers, six plants), WT (*n* = 38 flowers, two plants) and *5mCUC1* (*n* = 104 flowers, five plants). Petal number differs significantly between genotypes (Welch one-way ANOVA, *p* = 3.56e−103), and different letters denote statistical significance at *p* < 0.05 using a Games–Howell pairwise comparison as post hoc analysis. (c-d) Representative floral primordia surface reconstructions of *C. hirsuta 2x35S::miR164b* (*n* = 9), WT (*n* = 30) and *5mCUC1* (*n* = 7) (top and side view projections) showing cell area extension (heat map:cell growth ratio) during 10 h of growth (c) and *DR5v2::NLS:3xVenus* signal (heat map:average epidermal cell signal intensity in arbitrary units) at floral stage 4 (d). Arrows in (c) point to one inter-sepal boundary in each image, in top and side views. White dashed circles in (d) indicate DR5 expression maxima in inter-sepal boundaries. (e-h) Radial quantification of DR5 signal (sum of epidermal signal intensity in arbitrary units) in stage 4 floral primordia of *C. hirsuta* WT (e, f), *2x35S::miR164b* (g) and *5mCUC1* (h). In grey: 95% confidence interval. Pink shading: petal regions at approximately 45°, 135°, 225° and 315°. Plots in (e, g, h) correspond to the samples shown in (d), and in (f) to the mean of 10 WT samples. Scale bars: 2 mm (a: inflorescences), 1 mm (a: flowers), 20 μm (c, d).

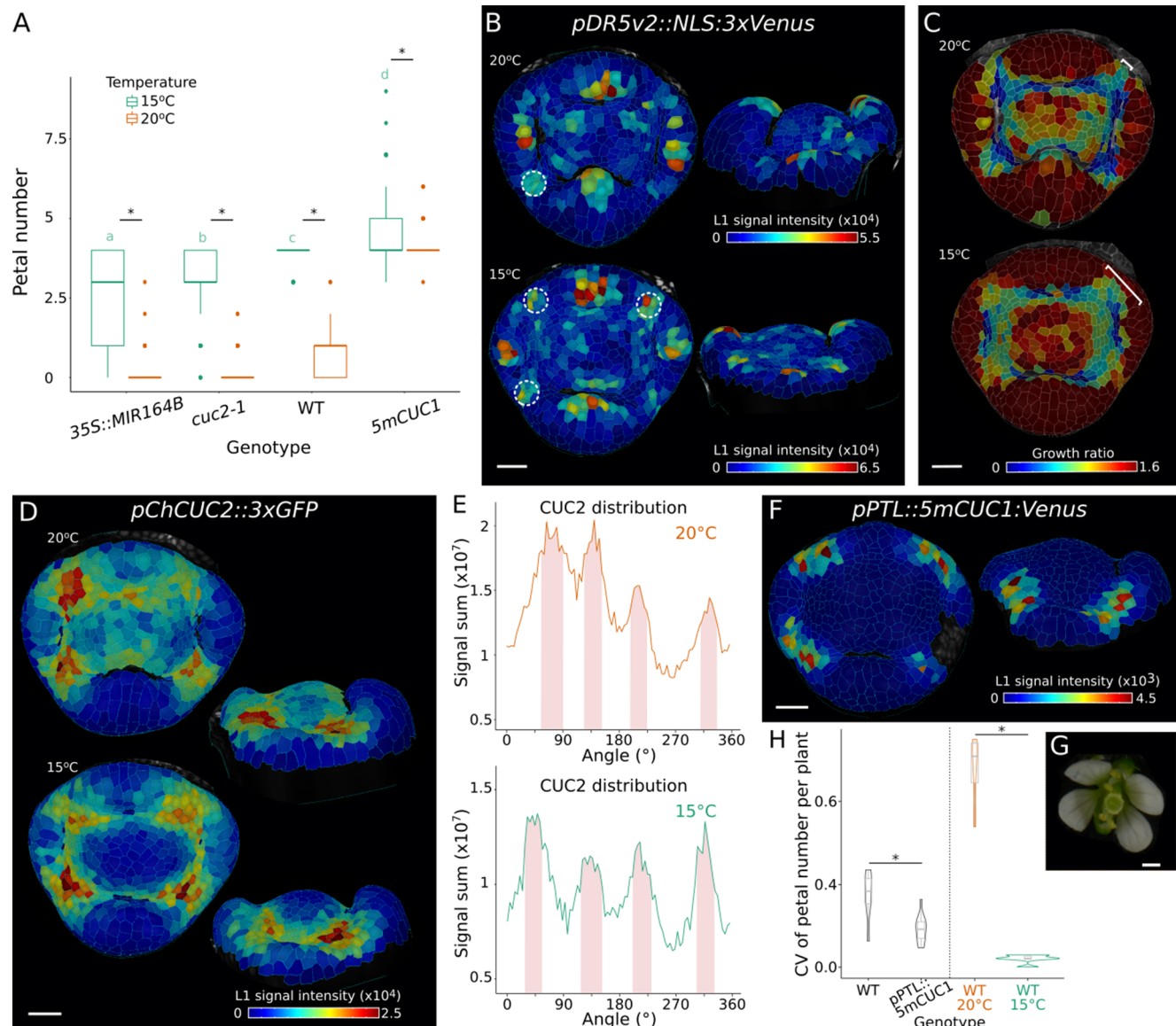

**Figure 2.** Ambient temperature regulates petal number independent of CUC activity in *Cardamine hirsuta*. (a) Boxplots of average petal number in *C. hirsuta 2x35S::miR164b*, *cuc2-1*, WT and *5mCUC1* grown at 15°C (green) and 20°C (orange). At 15°C, *2x35S::miR164b* (*n* = 187 flowers from 8 plants), *cuc2-1* (*n* = 150 flowers from 6 plants), WT (*n* = 150 flowers from 6 plants), *5mCUC1* (*n* = 242 flowers from 10 plants); at 20°C the same data as in Figure 1b was used. Two-way ANOVA shows that petal number differs significantly between genotypes (*, *p* = 2.04e−223), growth temperatures (*p* = 4.56e−178) and genotype:temperature interactions (*p* = 1.90e−78); different letters denote statistical significance at *p* < 0.05 for genotypes grown at 15°C using Tukey's HSD test as post hoc analysis. (b, d) Expression of *pDR5v2::NLS:3xVenus* (b, *n* = 11 at 20°C, *n* = 6 at 15°C) and *pChCUC2::3xGFP* (d, *n* = 2 at 20°C, *n* = 3 at 15°C) in representative floral primordia of *C. hirsuta* grown at 20°C and 15°C (top and side view projections). Heat maps show average epidermal cell signal intensity in arbitrary units. White dashed circles in (b) indicate DR5 expression maxima in inter-sepal boundaries. (c) Maps of cell area extension on representative floral primordia surface reconstructions of *C. hirsuta* WT grown at 20°C and 15°C (heat map: cell growth ratio, *n* = 30 at 20°C, *n* = 17 at 15°C) during 14 and 21 h of growth respectively. Brackets indicate the width of one inter-sepal boundary in each image. (e) Radial quantification of *ChCUC2* signal in samples shown in (d). Pink shading: petal regions at approximately 45°, 135°, 225° and 315°. (f) Expression of *pPTL::5mCUC1:Venus* in representative floral primordia of *C. hirsuta* (*n* = 2). Heat maps show average epidermal cell signal intensity in arbitrary units. (g) Representative flower of *C. hirsuta* expressing *pPTL::5mCUC1:Venus*. (h) Violin plots of coefficients of variation (CV) of petal number per plant in *C. hirsuta pPTL::5mCUC1:Venus* (*n* = 330 flowers, 15 plants) and in the corresponding WT (*n* = 128 flowers, 6 plants) and in WT grown at 20°C (*n* = 78 flowers, 4 plants) or 15°C (*n* = 150 flowers, 6 plants). CV differs significantly between genotypes (Wilcoxon test: WT-*pPTL::5mCUC1:Venus p* = 0.0154; WT20°C–WT15°C *p* = 2.02e−46), ∗ statistical significance at *p* < 0.05. Scale bars: 20 μm (b, c, d, f), 0.5 mm (g).

Therefore, both proteins are expressed at the right time and place to potentially contribute to petal initiation in *C. hirsuta*, and increased ChPIN1 expression was associated with the increased petal number observed at 15°C.

Since PIN1 proteins transport auxin in a polar manner (Galweiler et al., 1998), we analysed the polarity of the plasma membrane localisation of ChPIN1 in inter-sepal boundary cells of the same stage 4 flowers described earlier (Figure 3g,h).

We computed the direction of ChPIN1 polarity in each cell as previously described (Hu et al., 2024), and indicated this direction using arrows to point to the most intense signal. Coordinated fields of ChPIN1 polarities pointed to the direction of sepal tips in all flowers (magenta arrows, Figure 3g–j and Supplementary Figure S2). Between these fields, we observed divergent ChPIN1 polarities (white arrows, Figure 3g–j) that occasionally formed convergence points (asterisks, Figure 3g–j). Such convergence

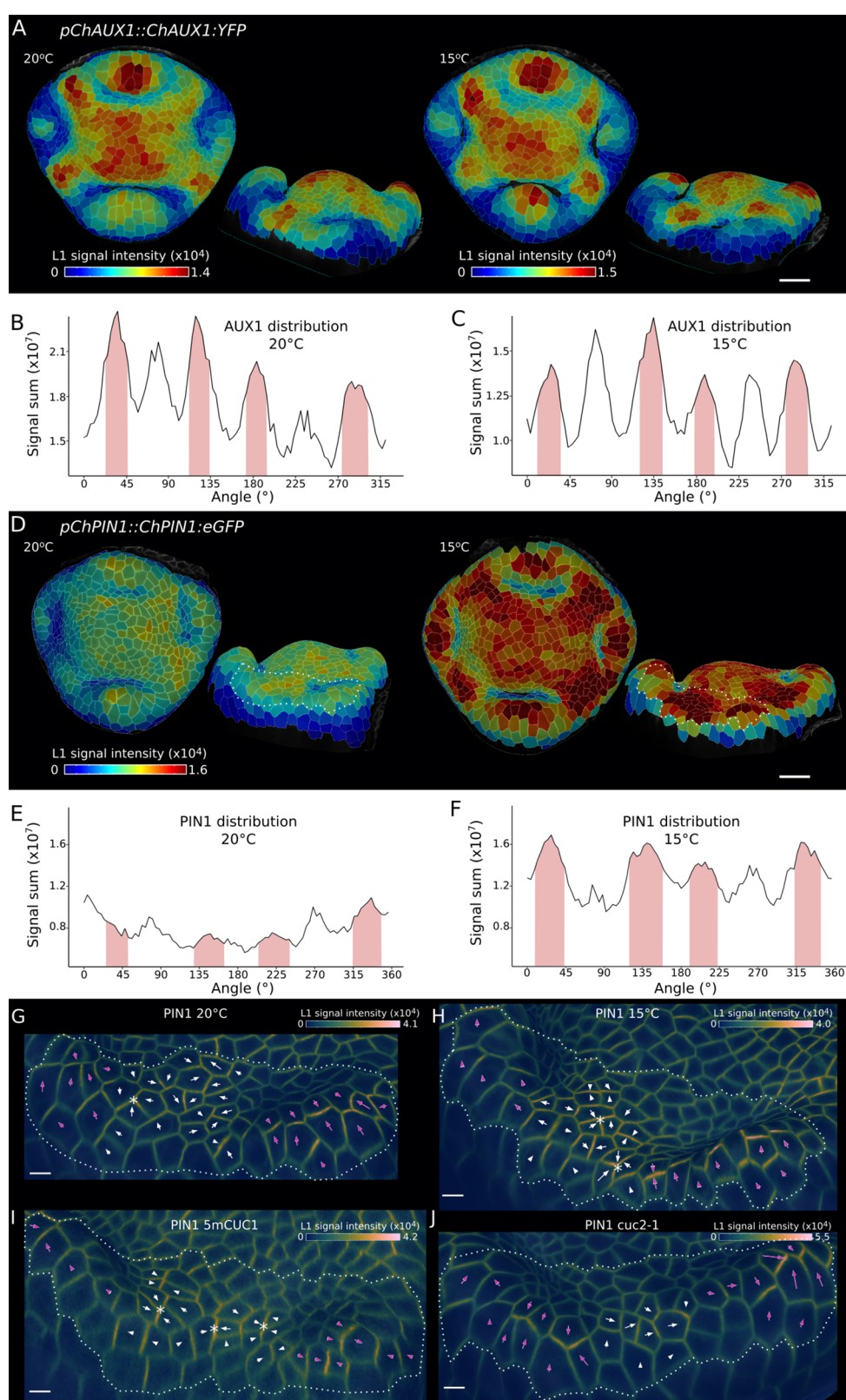

**Figure 3.** AUX1 and PIN1 localise to petal initiation regions in *Cardamine hirsuta* flowers. (a, d) *pChAUX1::ChAUX1:YFP116* (a) and *pChPIN1::ChPIN1:eGFP* (d) expression in representative floral primordia of *C. hirsuta* grown at 20°C (a: *n* = 1, d: *n* = 5) and 15°C (a: *n* = 4, d: *n* = 4), shown in top and side view projections. Heat maps show average epidermal cell signal intensity in arbitrary units. Segmentations based on AUX1 signal at the plasma membrane rather than PI staining in (a). White dashed lines in (d) outline the inter-sepal regions shown in (g) and (h). (b-c, e-f) Radial quantification of *pChAUX1::ChAUX1:YFP116* (b, c) and *pChPIN1::ChPIN1:eGFP* (e, f) signal (sum of epidermal signal intensity in arbitrary units) in *C. hirsuta* grown at 20°C (b, e) and 15°C (c, f). Petal regions are indicated on plots at approximately 45°, 135°, 225° and 315° (pink shading). (g-j) Plasma membrane-localised signal of *pChPIN1::ChPIN1:eGFP* in inter-sepal regions of representative *C. hirsuta* samples grown at 20°C (g, *n* = 5) and 15°C (h, *n* = 4), and *5mCUC1* (i, *n* = 2) and *cuc2-1* (j, *n* = 1) grown at 20°C (side view projections). Heat maps show epidermal signal intensity at the cell edges in arbitrary units. Arrows estimate cell polarity of the signal using vectors computed by MGX with arrowheads added manually to face the most intense signal. Pink arrows are in sepal domains and white ones are in inter-sepal domains. Scale bars: 20 μm (a, d), 5 μm (g–j).

points formed more frequently between sepals in wild-type flowers grown at 15°C compared to 20°C (asterisks, 15/16 boundaries formed convergence points at 15°C, Figure 3h, vs. 7/16 at 20°C, Figure 3g and Supplementary Figure S2). We also observed a higher frequency of ChPIN1 convergence points in the large inter-sepal regions of *5mCUC1* flowers (asterisks, Figure 3i, 8/8 boundaries formed convergence points, Supplementary Figure S2) and a lower frequency between sepals in *cuc2-1* flowers (Figure 3j, 1/4 boundaries formed convergence points). Since auxin maxima are formed by the convergence of PIN1 polarities (Heisler et al., 2005), these observations match the higher incidence of DR5 maxima between sepals in *5mCUC1* flowers (Figure 1d) and wild-type flowers with higher petal number at 15°C (Figure 2b). However, the observed ChPIN1 convergences in flowers with low petal number (Figure 3g,j) suggest that formation of a PIN1 convergence is not sufficient per se to form a stable peak of auxin activity that leads to petal initiation. In summary, higher petal number in *5mCUC1* and 15°C grown flowers was associated with more frequent formation of PIN1 convergence points and auxin maxima in larger inter-sepal boundaries.

### 2.4. Robust petal number in Arabidopsis

In comparison to *C. hirsuta*, petal number is canalised in Arabidopsis. To compare the localisation of *CUC2*, auxin activity maxima, PIN1 and AUX1 in this robust patterning system, we analysed stage 4 flowers of Arabidopsis. At this stage, the cellular signal of *pAtAUX1::AtAUX1:YFP116* (AtAUX1) (Swarup et al., 2004) was intense in inter-sepal boundaries, as previously reported (Supplementary Figure S3) (Lampugnani et al., 2013). An Arabidopsis *pAtCUC2::nls:tdTomato* (AtCUC2) transcriptional reporter was expressed in inter-sepal boundaries and excluded from sepals (Figure 4a). In comparison to *C. hirsuta*, petal initiation sites marked by expression of the auxin activity reporter *DR5v2::nls:3xVenus* (DR5) were located on the floral meristem (Figure 4b). In this way, no overlap occurred between *AtCUC2* transcription in inter-sepal boundaries and sites of petal initiation. We next investigated the expression and polarity of Arabidopsis *pAtPIN1::AtPIN1:GFP* (AtPIN1) associated with the formation of DR5 peaks on the floral meristem. We found the highest AtPIN1 signal located throughout the sepal whorl (Figure 4c). Coordinated fields of AtPIN1 polarities pointed to the direction of sepal tips (white arrows, Figure 4d). Immediately adjacent to the inter-sepal domains, we observed additional fields of AtPIN1 polarity running to the direction of the floral meristem, and therefore towards sites of petal initiation (grey arrows, Figure 4d).

Previous work had established that auxin is a mobile petal initiation signal in Arabidopsis (Lampugnani et al., 2013). It had been shown that increasing auxin biosynthesis in the inter-sepal domain in *ptl* mutants was sufficient to restore petal initiation in the adjacent whorl on the floral meristem (Lampugnani et al., 2013). Therefore, we investigated the localisation of AtPIN1 in the *ptl-1* allele, which shows almost complete petal loss (Figure 4e,f). As previously described, the loss of petals in *ptl-1* flowers was associated with the absence of DR5 maxima at petal initiation sites on the floral meristem (Figure 4g) (Lampugnani et al., 2013). The AtPIN1 signal was reduced throughout the sepal whorl in *ptl-1* compared to wild type and conspicuously low in the adjacent whorl where petals would normally form (Figure 4h). Coordinated fields of AtPIN1 polarities were visible in the sepal whorl, pointing to the direction of sepal tips (white arrows, Figure 4i). However, AtPIN1 signal was not visible in the adjacent whorl (Figure 4i). Therefore, *ptl-1* did not

show the fields of AtPIN1 polarity observed in wild-type flowers that putatively transport auxin from inter-sepal regions to sites of petal initiation on the floral meristem (Figure 4i).

Previous studies have used the bacterial auxin biosynthesis gene *iaaH* under the control of the *PTL* promoter to restore petal initiation in *ptl* flowers (Lampugnani et al., 2013). These experiments demonstrated the sufficiency of local auxin biosynthesis for petal initiation. However, it was not clear where endogenous auxin biosynthesis occurred in developing flowers. To investigate this question, we analysed *YUCCA* (*YUC*) genes, which are involved in the main auxin biosynthesis pathway (Cheng et al., 2006; 2007; Stepanova et al., 2008). Quadruple *yuc1 yuc2 yuc4 yuc6* mutants show strong floral defects, including loss of petals (Figure 5a) (Cheng et al., 2006). To assess the contribution of individual *YUC* genes to petal formation, we first imaged transcriptional reporters for *YUC1*, *YUC4* (Zhang et al., 2020) and *YUC2*, *YUC6* (Galvan-Ampudia et al., 2020). We failed to detect *pYUC6::GFP* expression in young flowers and found only faint expression of *pYUC2::GFP* in internal floral tissues near the pedicel (Supplementary Figure S4). In contrast to this, *pYUC1::NLS:3xGFP* (*YUC1*) expression was confined to a very specific domain in stage 4 flowers, consisting of a few cells between sepals (Figure 5c). At this stage, *pYUC4::NLS:3xGFP* (*YUC4*) was expressed more broadly in the floral meristem and excluded from the sepal whorl (Figure 5d). We observed high *YUC4* expression adjacent to inter-sepal zones at sites of petal initiation on the floral meristem (Figure 5d). To assess whether these two genes regulate petal number, we analysed single and double *yuc1 yuc4* mutants. We found no difference in petal number between wild-type and *yuc1*; a small, but significant reduction of petal number in *yuc4* and almost complete loss of petals in *yuc1 yuc4* double mutants (Figure 5b). Therefore, *YUC1* and *YUC4* act redundantly to control petal initiation. The expression domains of *YUC1* and *YUC4* suggest that two sites of local auxin production may be relevant for petal initiation – one defined by *YUC1* expression in the inter-sepal boundary, and another site in the adjacent whorl defined by *YUC4* expression. Therefore, based on genetic analysis and gene expression patterns, it is likely that *YUC1* and *YUC4* contribute to local auxin synthesis during petal initiation in developing flowers.

In summary, invariant petal number in Arabidopsis is associated with clear patterning by *CUC2* expression defining inter-sepal boundaries and the formation of auxin maxima in the adjacent whorl on the floral meristem. Auxin promotes petal initiation at these floral meristem sites in Arabidopsis (Lampugnani et al., 2013), and here we implicate polar auxin transport by PIN1, and local auxin biosynthesis by *YUC1* and *YUC4* in this process.

## 3. Discussion

Our analysis shows that *CUC1,2* genes regulate petal initiation in *C. hirsuta* by creating boundaries of reduced cellular growth between sepals and promoting the convergence of PIN1 polarities to create auxin maxima in these boundaries. This positioning of auxin maxima between sepals, rather than in the adjacent whorl on the floral meristem, distinguishes petal formation in *C. hirsuta* from Arabidopsis. In *C. hirsuta*, the space available between sepals for discrete auxin maxima to form is critical for petal initiation. In this way, growth of the inter-sepal region is an integral part of the patterning process controlling petal initiation, and subject to variation by environmental and genetic factors that cause petal

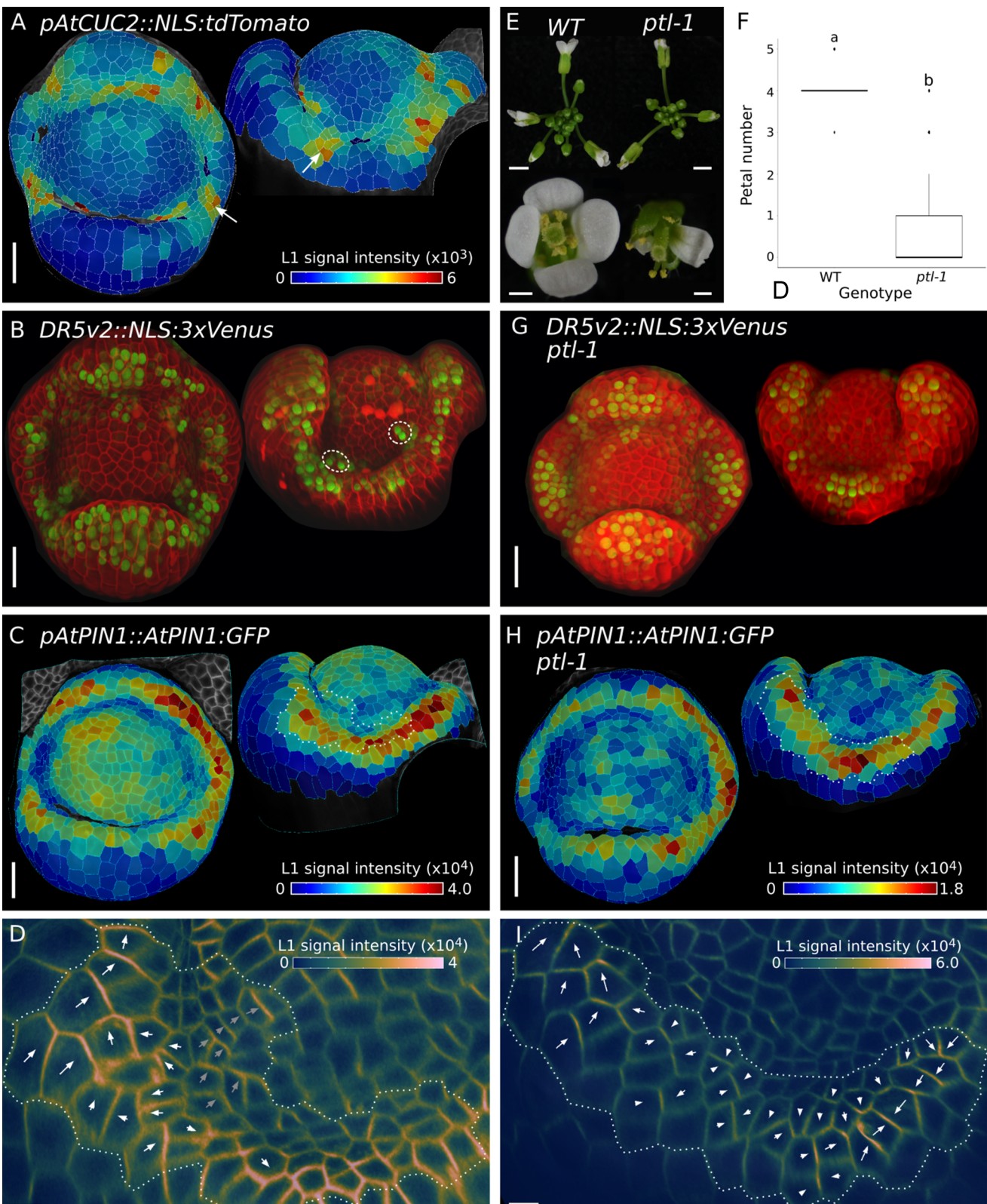

**Figure 4.** Robust petal number in Arabidopsis. (a, c, h) Heat maps quantifying the epidermal signal in representative samples of *pAtCUC2::NLS:tdTomato* in WT (a, *n* = 10) and *pAtPIN1::AtPIN1:GFP* in WT (c, *n* = 4) and *ptl-1* (h, *n* = 2) stage 4 floral primordia of *Arabidopsis thaliana*. Colour bars: epidermal signal intensity in arbitrary units. Arrows in (a) point to *AtCUC2* expression in one inter-sepal boundary in top and side views. White dashed lines in (c) and (h) outline the inter-sepal regions shown in (d) and (i). (b, g) *pDR5v2::NLS:3xVenus* (green) expression in representative samples of WT (b, *n* = 6) and *ptl-1* (g, *n* = 3) Arabidopsis flowers at stage 4; cells outlined with PI staining (red). Top and side views are shown, white dashed circles in side view (b) indicate DR5 expression maxima on the flanks of the floral meristem. (d, i) Projections of plasma membrane-localised signal of *pAtPIN1::AtPIN1:GFP* in inter-sepal regions of Arabidopsis WT (d) and *ptl-1* (i), shown as side views. Heat maps show epidermal signal intensity at the cell edges in arbitrary units. Arrows estimate cell polarity of the signal using vectors computed by MGX with arrowheads added manually to face the most intense signal. White arrows point to sepal tips, grey arrows point to FM. (e) Representative inflorescences (top) and flowers (bottom) of Arabidopsis WT and *ptl-1*. (f) Boxplots of petal number in Arabidopsis WT (*n* = 150 flowers, six plants) and *ptl-1* (*n* = 150 flowers, six plants). Letters denote statistically significant differences between means based on Welch one-way ANOVA (*p* = 1.68e−80). Scale bars: 20 μm (a, b, c, g, h), 5 μm (d, i), 2 mm (inflorescences) and 0.5 mm (flowers) (e). Image in (b) reproduced from Monniaux et al. (2018).

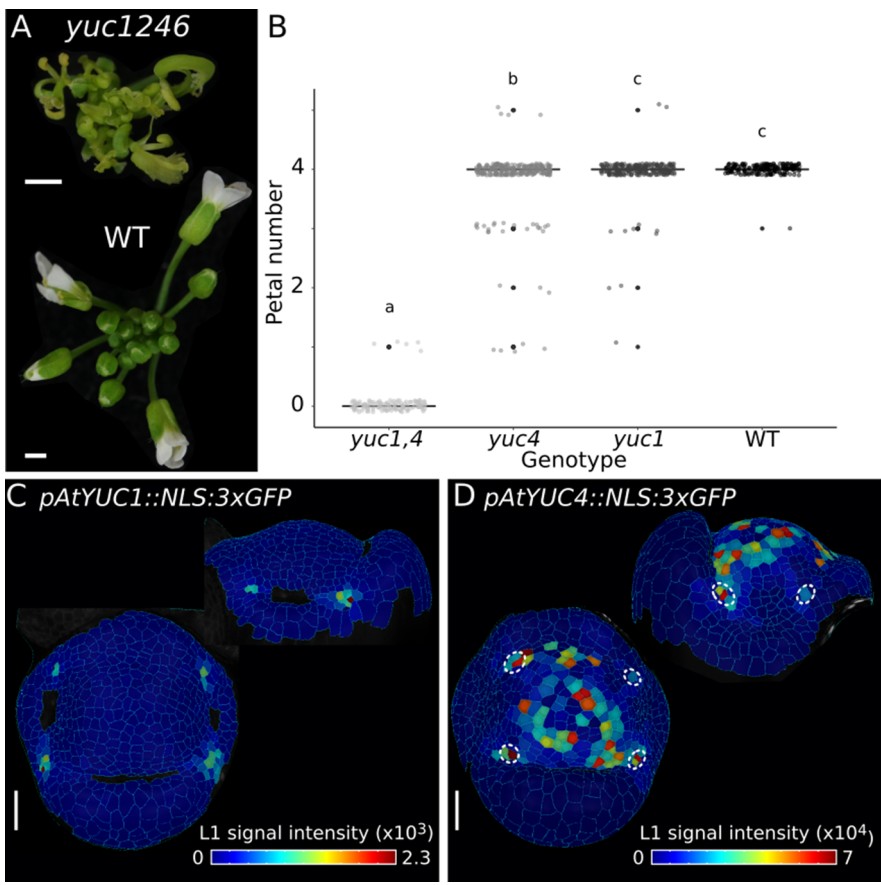

**Figure 5.** Auxin biosynthesis genes *YUC1* and *YUC4* contribute to petal number in Arabidopsis. (a) Representative inflorescences of Arabidopsis WT and *yuc1246* mutant. (b) Boxplots of petal number in *yuc1yuc4* (*n* = 137 flowers, 6 plants), *yuc4* (*n* = 278 flowers, 12 plants), *yuc1* (*n* = 261 flowers, 12 plants) and WT plants (*n* = 206 flowers, 9 plants). Letters denote statistically significant differences ($p < 0.05$) between means based on Welch one-way ANOVA ($p = 0$, Tukey's HSD post hoc test). *P* values are 0.659 (Col-0-*yuc1*), 0 (Col-0-*yuc1yuc4*), 3.88e−4 (Col-0-*yuc4*), 0 (*yuc1-yuc1yuc4*), 1.34e−2 (*yuc1-yuc4*) and 0 (*yuc1yuc4-yuc4*). (c-d) Heat maps quantifying epidermal signal in representative samples of *pAtYUC1::NLS:3xGFP* (c, *n* = 5) and *pAtYUC4::NLS:3xGFP* (d, *n* = 4) in Arabidopsis stage 4 flowers (top and side view projections). Colour bars: epidermal signal intensity in arbitrary units. White dashed circles in (d) indicate *YUC4* expression in petal initiation regions on the floral meristem. Scale bars: 1 mm (a), 20 μm (c, d).

number to vary in *C. hirsuta* (McKim et al., 2017; Pieper et al., 2016).

We have shown that increasing or decreasing CUC activity in *C. hirsuta* shifts the distribution of petal number to a higher or lower mean, respectively, than the wild type. This is different to the canalising effect of transferring *AP1* from Arabidopsis into *C. hirsuta*, which produces invariant petal number (Monniaux et al., 2018). The effect of this Arabidopsis *AP1* transgene is associated with an expanded expression domain that extends onto the floral meristem in *C. hirsuta*. In comparison, the endogenous domain of *C. hirsuta* *AP1* expression is restricted to the sepal whorl and petal initiation occurs between sepals (Monniaux et al., 2018). Yet, according to the 'ABC' model, petal formation requires not only *AP1*, but also 'B' genes, such as *APETALA3* (*AP3*) and *PISTILLATA*, which are excluded from the sepal whorl in Arabidopsis (Coen & Meyerowitz, 1991). We verified that a functional Arabidopsis AP3 fusion protein was restricted to the floral meristem in Arabidopsis, but extended out into inter-sepal regions in *C. hirsuta* flowers (Supplementary Figure S5), which matched the endogenous expression domain of *C. hirsuta* *AP3* (Supplementary Figure S5). Therefore, a potential overlap exists between *AP1* and *AP3* expression in the inter-sepal boundaries where petals initiate in developing *C. hirsuta* flowers.

Our findings indicate that *CUC* genes play a dual role in promoting the competence of inter-sepal regions to give rise to petals

in *C. hirsuta*. First, *CUC* genes create boundary regions of reduced cellular growth that maintain the competence to proliferate (Bhatia et al., 2023; Vernoux et al., 2000). This is a general feature of CUC action; for example, during early leaf development in *C. hirsuta*, CUC2 limits the growth, but not proliferation, of cell populations along the leaf margin to pattern leaflet boundaries (Bhatia et al., 2023). Second, CUC activity promotes the convergence of PIN1 polarities, which are responsible for creating auxin maxima. For example, CUC1 acts in *C. hirsuta* leaves to repolarise PIN1 via the direct transcriptional activation of WAG kinases that phosphorylate PIN1 (Hu et al., 2024). Testing whether CUC1 regulates PIN1 polarity via the same mechanism in *C. hirsuta* flowers will be an important follow-up to this study.

The positioning of auxin maxima in inter-sepal boundaries is likely to create conflicts that impair petal initiation in *C. hirsuta*. On the one hand, auxin maxima trigger organogenesis, but on the other hand, CUC activity specifies boundaries that should be devoid of organ formation. Similar conflicts were proposed to contribute to the failure of *pin1* inflorescences to initiate flowers. Naked *pin1* apices were shown to express early markers of organ initiation, such as *AINTEGUMENTA* and *LEAFY*, together with *CUC2*, creating a hybrid organ/boundary identity at the meristem periphery (Vernoux et al., 2000). We found that petal initiation was more likely to succeed in *C. hirsuta* when larger boundary

regions were created between sepals, either by slow growth and prolonged maturation at low ambient temperatures or increased CUC1 activity in these regions. We observed the convergence of PIN1 polarities more frequently and more regular spatial patterning of auxin maxima in these larger inter-sepal regions. This suggests that organ versus boundary identities may have the space to resolve with less overlap in a larger tissue context. Moreover, we found that the effect of low ambient temperature to increase petal number and reduce its variability was independent of *CUC1,2* gene activity. Therefore, additional factors that contribute to auxin patterning may have more space to resolve in larger inter-sepal regions.

A key feature of robust petal initiation in Arabidopsis is the formation of auxin maxima on the floral meristem. It was previously shown that auxin is a mobile signal for petal initiation and that factors in inter-sepal regions, such as AUX1 and PTL, influence the availability of auxin to accumulate in the adjacent whorl (Lampugnani et al., 2013). Our findings indicate how polar transport and local synthesis of auxin may contribute to this process. We observed a divergent field of PIN1 polarity in Arabidopsis flowers that may provide a putative transport route for auxin from the inter-sepal region to the adjacent whorl on the floral meristem. This PIN1 localisation was absent from the floral meristem in *C. hirsuta* flowers and in *ptl-1* mutants that lacked petals. Based on previous work (Hu et al., 2024), we hypothesise that CUC may instruct the repolarisation of PIN1 in inter-sepal boundaries of Arabidopsis flowers to initiate new auxin fluxes towards petal initiation sites on the floral meristem. Testing this hypothesis may help to further understand how auxin controls robust petal initiation in Arabidopsis.

CUC activity has also been shown to influence the expression of *YUCCA* auxin biosynthetic genes (Abley et al., 2016; Yamada et al., 2022). Our genetic analysis indicates that *YUC1* and *YUC4* contribute to petal initiation in Arabidopsis. *YUC1* expression suggests that auxin is likely synthesised in inter-sepal regions, which may provide a local source of auxin for polar transport to petal initiation sites via PIN1. Furthermore, the expression of *YUC1* in inter-sepal regions of developing flowers also matched previous reports for the auxin biosynthetic genes *TRYPTOPHAN AMINOTRANSFERASE OF ARABIDOPSIS 1* (*TAA1*) and *TAR2* (Yadav et al., 2023). On the other hand, *YUC4* expression suggests that auxin is also likely to be synthesised locally at petal initiation sites on the floral meristem. Therefore, auxin synthesised at a distance in inter-sepal regions and also locally on the floral meristem are both likely to contribute to the formation of auxin maxima for petal initiation in Arabidopsis flowers. Comparing the function and distribution of *YUCCA* genes in *C. hirsuta* flowers may help to further understand the variable initiation of petals in this species.

In summary, decanalised petal number in *C. hirsuta* is associated with a shift in petal initiation events from the floral meristem to inter-sepal boundaries. This is underpinned by *AP1* gene divergence in *C. hirsuta*, which resulted in loss of *AP1* expression from the floral meristem and its loss of epistasis over petal number QTL (Monniaux et al., 2018). Attempting to initiate petals in boundary regions likely exposed this developmental process to the influence of many genetic loci that are otherwise cryptic in a robust system. Identifying the genes underlying petal number QTL (Monniaux et al., 2016; 2018; Pieper et al., 2016), and studying the natural variation and genetic basis for petal number plasticity in response to temperature, will be important avenues of future research to gain insight into these questions.

## 4. Materials and methods

### 4.1. Accessions and plant material

**Table 1.** Plant material used in this study.

| Species | Designation | Source or reference |
|---|---|---|
| *C. hirsuta* | Oxford (WT accession) | Hay and Tsiantis (2006) |
| *A. thaliana* | Col–0 (WT accession) | CS60000 |
| *C. hirsuta* | 2x35S::MIR164B | Blein et al. (2010) |
| *C. hirsuta* | cuc2–1 | Rast-Somssich et al. (2015) |
| *C. hirsuta* | pCUC1::5mCUC1 | This article; plasmid from Mallory et al. (2004) |
| *C. hirsuta* | pChCUC2::3xGFP:3′ | This article |
| *C. hirsuta* | pDR5v2::NLS:3xVenus | Monniaux et al. (2018) |
| *C. hirsuta* | pChPIN1::ChPIN1:eGFP | Hu et al. (2024) |
| *C. hirsuta* | pAtPTL::5mCUC1:Venus | This article |
| *C. hirsuta* | pChAUX1::ChAUX1:YFP116 | This article |
| *C. hirsuta* | pChAP3::ChAP3:GFP | This article |
| *C. hirsuta* | pAtAP3::AtAP3:GFP | This article |
| *A. thaliana* | pAtAP3::AtAP3:GFP in ap3–7 | This article |
| *A. thaliana* | pAtAUX1::AtAUX1:YFP116 | Swarup et al. (2004) |
| *A. thaliana* | pAtCUC2::NLS:tdTomato | This article |
| *A. thaliana* | pDR5v2::NLS:3xVenus | Liao et al. (2015) |
| *A. thaliana* | pAtYUC1::NLS:3xGFP | Zhang et al. (2020) |
| *A. thaliana* | pYUC2::GFP | Galvan-Ampudia et al. (2020) |
| *A. thaliana* | pAtYUC4::NLS:3xGFP | Zhang et al. (2020) |
| *A. thaliana* | pYUC6::GFP | Galvan-Ampudia et al. (2020) |
| *A. thaliana* | pAtPIN1::AtPIN1:GFP | Heisler et al. (2005) |
| *A. thaliana* | yuc1 | SALK_106293 |
| *A. thaliana* | yuc4–1 | SM_3_16128 |
| *A. thaliana* | ptl–1 | N276; Griffith et al. (1999) |
| *A. thaliana* | ap3–7 | This article; CS6565 |

Plant material and accessions used in this study are described in Table 1. *ap3-7* is a Col-0 allele identified in seed stock CS6565 with a GT>GA mutation in the splice site of intron 4. The described mutation for *ap3-6* was not present in this stock. Therefore, we named the allele *ap3-7*. Genotyping primers f324/f325 were used to amplify a 342-bp fragment that is cut by MboI in the *ap3-7* mutant, and remains uncut in the wild type. *yuc4-1 yuc1* double mutants were identified from selfed *yuc4-1 yuc1/+* plants using genotyping primers lr063/lr064, lr066/lr067, LBb1.1 and Spm32.

### 4.2. Transgenic plant construction

The wild-type genotype in *C. hirsuta* is the reference Oxford (Ox) accession, herbarium specimen voucher Hay 1 (OXF) (Hay & Tsiantis, 2006) and in Arabidopsis, the reference Col-0 accession. All plasmids were transformed into wild-type plants by *Agrobacterium tumefaciens* (strains GV3101 or C58)-mediated floral dip (Clough & Bent, 1998).

The *pChAUX1::ChAUX1:YFP116* plasmid contains an in-frame fusion of YFP after amino acid 116 of *ChAUX1*, as described for functional Arabidopsis AUX1-YFP fusions (Swarup et al., 2004).

A 3.4-kb *ChAUX1* promoter sequence (3.4 kb before ATG) was amplified from *C. hirsuta* genomic DNA with primers m272/m434, digested with *Sal*I/*Xma*I and subcloned into pBJ36. A 450-bp sequence of the 5ʹ end of *gChAUX1* (starting at ATG) was amplified with primers m276/m430, and YFP coding sequence (715 bp) was amplified with primers m431/m432. Both fragments were fused by overlapping PCR so that YFP was inserted after amino acid 116 of *gChAUX1*. This fragment was fused by overlapping PCR with a 2.8-kb sequence of the 3ʹ end of *gChAUX1* (amplified with primers m433/m277). The whole fragment was subcloned into pBJ36 by *Xma*I/*Bam*HI digestion and validated by sequencing. Functionality of the construct was tested by complementing Arabidopsis *aux1-22* mutants: five independent lines had a single transgene copy (IDna Genetics, Norwich, UK) and rescued the root gravitropism phenotype of *aux1-22* seedlings. This functional construct was transformed into *C. hirsuta*.

The following three constructs were generated by GreenGate cloning (Lampropoulos et al., 2013). *pAtPTL::5mCUC1:Venus*: a 2.8-kb *pAtPTL* promoter sequence (consisting of 1.3 kb before the ATG, the 5ʹUTR, the first exon and intron and eight codons of the second exon), that drives functional *PTL* expression (Lampugnani et al., 2013), was amplified from Arabidopsis genomic DNA. *5mCUC1* was amplified from *pCUC1::5mCUC1* plasmid (Mallory et al., 2004). Entry vectors were cloned into the binary vector pGGZwf01 (Perez-Anton et al., 2022). *pAtAP3::AtAP3:GFP*: a 696-bp *AtAP3* sequence was amplified from Arabidopsis cDNA as two fragments with primers m478/m472 and m476/m479 to mutate an internal *Bsa*I site. This entry vector was cloned with pGGA002 (Addgene plasmid #48813; Lampropoulos et al., 2013), which contains a 1.269-kb *pAtAP3* promoter sequence, into the binary vector pGGZ003. Fifteen independent lines were generated in *C. hirsuta* and 24 in Arabidopsis *ap3-7*. Lines that complemented the sterility of homozygous *ap3-7* mutants were analysed for transgene copy number (IDna Genetics, Norwich, UK) and three lines with either one or two transgene copies were imaged in T2 and T3 plants. *pChAP3::ChAP3:GFP*: a 927-bp *pChAP3* promoter sequence was amplified from *C. hirsuta* genomic DNA with primers m469/m470. A 696-bp *ChAP3* sequence was amplified from *C. hirsuta* cDNA as two pieces with primers m480/m475 and m476/m481 to remove an internal *Bsa*I site. Entry vectors were cloned into the binary vector pGGZ003. Functionality of the construct was tested by complementing Arabidopsis *ap3-7* mutants. This functional construct was transformed into *C. hirsuta*, six independent lines were analysed for transgene copy number (IDna Genetics, Norwich, UK) and three lines with either one or two transgene copies were imaged in T2 and T3 plants.

The *pChCUC2::3xGFP:3ʹ* plasmid contains 2.6-kb *C. hirsuta CUC2* promoter sequence (2.6 kb before ATG) and 2.6-kb 3ʹ sequence (2.6 kb after STOP) amplified from *C. hirsuta* genomic DNA and fused with 3xGFP. Fragments were subcloned into pBS KS+ as follows: 2.6-kb 5ʹ*CUC2* promoter in *Sac*I and *Bam*HI sites, 3xGFP in *Bam*HI and *Pst*I sites, and 2.6-kb 3ʹ*CUC2* in *Pst*I and *Sal*I sites; then amplified using primers with *Not*I overhangs and cloned into pMLBART. The *pAtCUC2::NLS:tdTomato* reporter was generated by fusing a 3.2-kb Arabidopsis *CUC2* promoter sequence (3.2 kb before ATG), amplified from *pAtCUC2::AtCUC2:Venus* plasmid (Heisler et al., 2005), to tdTomato and cloned into pMLBART-hyg.

All primers used in this study are listed in Table 2.

## 4.3. List of primers used.

**Table 2.** Primers used in this study.

| Primer | 5ʹ→3ʹ sequence | Experiment |
| --- | --- | --- |
| lr063 | ACCATATGAGGCAGAGCATTG | Genotyping SM_3_16128 (*yuc4–1*) |
| lr064 | ATCTCCACCATTTTTCCCTTC | Genotyping SM_3_16128 (*yuc4–1*) |
| lr066 | GGTTCATGTGTTGCCAAGGGA | Genotyping SALK_106293 (*yuc1*) |
| lr067 | CCTGAAGCCAAGTAGGCACGTT | Genotyping SALK_106293 (*yuc1*) |
| LBb1.3 | ATTTTGCCGATTTCGGAAC | Genotyping SALK tDNA insertions |
| Spm32 | TACGAATAAGAGCGTCCATTTTAGAGTGA | Genotyping SM tDNA insertions |
| m272 | AAACGTGTCGACTAGAGGTAGGTAAGGTCTCG | Cloning *ChAUX1* promoter (3.4 kb) |
| m434 | AAACGTCCCGGGTTTTTTTTAGATCTGAAAACAAAACGACC | Cloning *ChAUX1* promoter (3.4 kb) |
| m276 | AAACGTCCCGGGATGTCGGAAGGAGTAGAAGC | Cloning 5ʹ end of *ChAUX1* (450 bp) |
| m430 | AGCTCCTCGCCCTTGCTCACTTTGAAGCTTTTGCCTTCTTTCTC | Cloning 5ʹ end of *ChAUX1* (450 bp) |
| m431 | GAGAAAGAAGGCAAAAGCTTCAAAGTGAGCAAGGGCGAGGAGCT | Cloning YFP (715 bp) |
| m432 | GATAAGAAACCTGAATAACGTGGTTCTTGTACAGCTCGTCCATGC | Cloning YFP (715 bp) |
| m433 | GCATGGACGAGCTGTACAAGAACCACGTTATTCAGGTTTCTTATC | Cloning 3ʹ end of *ChAUX1* (2.8 kb) |
| m277 | AAACGTGGATCCTCAAAGAAGGTGGTGTAAAGCC | Cloning 3ʹ end of *ChAUX1* (2.8 kb) |
| f324 | GATCAAGTATTTGTTTCTCTCTCTTCTCTT | Genotyping *ap3–7* |
| f325 | AACATATATTATCCAACAGTAAACAAAATG | Genotyping *ap3–7* |
| m469 | GGTCTCAACCTAAAGTTTTAGTAGCCTTGTACGCT | Cloning *pChAP3* |
| m470 | GGTCTCATGTTGATATTTTTCTTTCTTCTCTATTTTTTTTA | Cloning *pChAP3* |
| m472 | GGTCTCTTTTCGATCTGATTCCCAAGAGATTTGAACT | Cloning *AtAP3* CDS |
| m475 | GGTCTCTTTTCGATCTGATTCCCAAGGGATTTTATCTTA | Cloning *ChAP3* CDS |
| m476 | GGTCTCAGAAACCACCAAGAAAAAGAACAAAAGTCAAC | Cloning *ChAP3* and *AtAP3* CDS |
| m478 | GGTCTCAGGCTTAATGGCGAGAGGGAAGATCCA | Cloning *AtAP3* CDS |
| m479 | GGTCTCACTGATTCAAGAAGATGGAAGGTAATGATGTC | Cloning *AtAP3* CDS |
| m480 | GGTCTCAGGCTTAATGGCGAGAGGAAAGATCCAG | Cloning *ChAP3* CDS |
| m481 | GGTCTCACTGATTCAAGAAGGTGGAAGGTAATGATGT | Cloning *ChAP3* CDS |
| lr050 | CCAGCCTTTGTATTTTCCCGT | Genotyping SALK_093708 (*yuc6*) |
| lr051 | CCGGAAAAAGGGTTCTTGTCG | Genotyping SALK_093708 (*yuc6*) |
| lr061 | ATAACCCAATCCAAACTTGCC | Genotyping SALK_093708 (*yuc6*) |
| f564 | TTCTTGCATTTTCTCGCTCTACG | Genotyping SALK_030199 (*yuc2*) |

### 4.4. Plant growth conditions and petal number scoring

Seeds were sown on soil, stratified 10 days in the dark at $4°C$, germinated for 7 days in short days (SD) and transferred to long-day (LD) conditions in the greenhouse. SD: 8 h light ($22°C$), 16 h dark ($20°C$), LD: 16 h light ($22°C$), 8 h dark ($20°C$). Plants grown at $15°C$ were sown on soil, stratified 10 days in the dark at $4°C$ and transferred into a growth cabinet with LD conditions: 16 h light ($16°C$, 70% relative humidity), 8 h dark ($14°C$, 70% relative humidity). Petal number was scored every second day in open flowers that were removed one by one with tweezers from the primary shoot. The first 25 flowers on the main shoot were scored. Due to the seasonal variability of petal number in *C. hirsuta* (McKim et al., 2017), data presented in each petal number plot always come from a single batch of experiments.

### 4.5. Confocal laser scanning microscopy

Floral primordia were staged as previously described in Arabidopsis (Smyth et al., 1990) and *C. hirsuta* (McKim et al., 2017). Time-lapse imaging was performed using plants grown as described earlier. Upon bolting, primary inflorescence meristems were cut and dissected under water. Flowers were removed one by one until the inflorescence meristem and young floral primordia up to stage 3 were uncovered. Dissected shoots were transferred to ½ MS supplemented with 1.5% plant agar (Duchefa), 1% sucrose and 0.1% plant preservative medium (Plant Cell Technology), with pH adjusted to 5.7 with KOH. Shoots were oriented in order to lay the inflorescence meristem or the floral meristem of interest flat and fixed with a drop of warm 1% agarose. Cell walls were stained with 0.1% propidium iodide (PI, Sigma) for 5 min before each observation, then rinsed twice in $H_2O$ and immersed in $H_2O$ at least 10 min before observation in order to transiently stop growth that would impair the acquisition of large image stacks. Water-immersed meristems were imaged from the top, on the day of the dissection and 14, 24 and 38 h after the initial time point in order to image stages 3 to 6 of floral development. Between each acquisition, the samples were placed into a LD growth chamber (at standard $20°C$ temperature or at $15°C$ when required). Confocal imaging was performed using an upright Leica SP8 confocal laser scanning microscope equipped with a long working distance water immersion objective (HC FLUOTAR L 25x/0.95 W) (Leica) and HyD hybrid detectors (Leica). Excitation was achieved with OPS lasers at 488 nm for GFP, 514 nm for Venus and YFP and 552 nm for tdTomato and PI. The emission fluorescence was collected at 500–520 nm for GFP, 520–540 nm for Venus and YFP, 575–590 nm for tdTomato and 630–650 nm for PI. Unidirectional scans (16 or 12 bit) of a $1024 \times 1024$ pixel region of interest were acquired at 600 Hz with a pixel size of 150–200 nm and a z-step size of 500 nm. Line averaging was either absent or set to 2.

### 4.6. Quantitative image analysis

Confocal image stacks were analysed with MorphoGraphX software (Barbier de Reuille et al., 2015): image stacks were loaded and the sample surface was detected and reconstructed in a 2.5D object by generating a mesh. All top- and side-view snapshots of flower meristems show quantitative data projected on 2.5D meshes. To quantify gene expression, the epidermal (2–5 μm) signal was projected onto the reconstructed surface and a heat map of signal intensity (border or interior signal depending on the reporter) averaged per cell was generated.

To quantify signal intensities at different positions relative to the centre of the floral meristem, the stacks were manually aligned in MorphoGraphX in order to place the z-axis at the centre pointing upward, the x-axis representing the $0°$ position and pointing to the right. Angles increased counter clockwise, within the *xy* plane. All images were aligned the same way, with a lateral sepal at position $0°$, the adaxial sepal at $90°$, the second lateral sepal at $180°$ and the abaxial sepal at $270°$. The function 'Export Histogram Circular' (in Stack -> Analyses) was used with the following parameters: bin number: 90, voxel value threshold %: 1, voxel distance min (μm): 0, voxel distance max (μm): 50 and weight by volume: yes. Cell signal intensities were exported as a csv file and imported into R for statistical analysis and visualisation.

To quantify cell growth with MorphoGraphX, we tracked cell lineages between two consecutive time points, computed growth ratios and displayed the data as a colour map on the second time point. In order to capture floral stage 4, samples grown at $20°C$ were imaged three times over 24 h of development and samples grown at $15°C$ were imaged three times over 43 h of development. Consecutive time points corresponding to a 10-h interval (Figure 1c), a 14-h interval ($20°C$, Figure 2c) or a 21-h interval ($15°C$, Figure 2c) were analysed.

To quantify PIN1 polarity with MorphoGraphX, the signal projected on the surface was colour coded based on its intensity (for visualisation) and the process 'compute signal orientation' was used. The displayed output consisted in min and max signal intensity axes drawn inside each segmented cell. Screenshots of regions of interest were taken with and without these axes being displayed. ImageJ (version 1.51i or earlier) was used to manually draw the max axis in cells of interest and to assign an arrowhead towards the edge with the most PIN1 expression based on visual comparison between the two edges of the computed max axis.

### 4.7. Statistical analysis

Petal number was statistically analysed with R Studio version 1.3.1093 or earlier (R Team, 2020). Throughout the article, average values are mean ± SD and boxplots show quartiles (box and whiskers), median (bold line), outliers (dots). A Welch one-way ANOVA followed by a Games–Howell post hoc test for pairwise comparisons was used in Figure 1b. A two-way ANOVA followed by a Tukey's HSD test was used to test the influence of both temperature and genotype on petal number in Figure 2a. A Welch one-way ANOVA was used in Figure 4f and a one-way ANOVA followed by a Tukey's HSD test was used for pairwise comparisons in Figure 5b. The coefficient of variation was calculated as the ratio of the standard deviation to the mean in Figure 2h, and a Wilcoxon rank sum test was used to compare two independent groups of samples.

### 4.8. Photographs of inflorescences and flowers

The photographs of inflorescences and individual flowers were taken with a Canon compact camera equipped with a macro lens EF 50 mm 1:25 and a lens macro 1:1 (Canon).

**Open peer review.** To view the open peer review materials for this article, please visit http://doi.org/10.1017/qpb.2025.10015.

### Acknowledgements

We thank M. Tsiantis for comments and sharing materials; P. Laufs and T. Vernoux for sharing materials; R. Dello Ioio for generating *5mCUC1 C. hirsuta*

transgenic lines; W. Faigl for technical assistance; N. Bhatia and S. Strauss for advice and D. Wilson Sanchez for advice and sharing materials.

**Competing interest.** The authors declare none.

**Data availability statement.** Raw data, confocal stacks in tiff format as well as MorphoGraphX segmentation meshes are made available at: https://doi.org/10.17617/3.93

**Author contributions.** AH conceived and designed the study. LR-L performed experiments and analysed data. LR-L, MM, ZH and SM generated materials. AH and LR-L wrote the article.

**Funding statement.** This work was supported by the Deutsche Forschungsgemeinschaft FOR2581 on Plant Morphodynamics (A.H., grant number HA 6316/2-1). M.M. and S.M. were supported by the European Molecular Biology Organization Long Term Fellowships, S.M. by a National Science and Engineering Research Council of Canada Post-Doctoral Fellowship.

**Supplementary material.** The supplementary material for this article can be found at http://doi.org/10.1017/qpb.2025.10015.

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
