## [Reviewer Report · Review: CUC/auxin patterning of decanalised petal number in *Cardamine hirsuta*
— R0/PR2]

CUC/auxin patterning of decanalized petal number in Cardamine hirsuta

Rambaud-Lavigne Léa, Monniaux Marie, Hu Zi-Liang, McKim Sarah, Hay Angela

In this manuscript the authors provide evidences that the CUC/auxin patterning is involved in the decanalized petal number in cardamine hirsute a close relative to Arabidopsis thaliana. They show that the same patterning modules are deployed differently between Arabidopsis and Cardamine hirsuta resulting in loss of robustness in petal initiation in Cardamine.

Overall I think the quality of the writing and the figures are very good. This is an interesting paper demonstrating how CUC and auxin pathways are integrated during flower development in two different species. The work presented here relies on time-lapse imaging of flower primordia in order to measure growth rate and gene expression patterns. Developmental robustness is a critical feature for crop improvement specially in changing environments therefore I think that this work may be of interest for a wide readership. Here are my comments:

Major comment:

Throughout the paper, the authors used time-lapse experiment to draw their conclusions regarding petal initiation in Cardamine hirsuta. For each figure, an experiment is shown but there is no indication of the number of biological replicates. Can the author at least provides the numbers? Time-lapse acquisition represent valuable data to understand developmental processes and are very challenging to obtain. As they did for PIN1 polarity in response to temperature, I think it could be important to show that the data presented are not the result of a single experiment but are representative of several replicates.

All along the paper, I feel that some statistical analyses could convinced the reader for instance when they explore how ChCUC2 expression varies between 15°C and 20°C (Fig2D-E) or, on Fig1C, when comparing growth rates in inter-sepal boundaries in various genotypes (not clear from the data presented).

Minor comments:

- line 74: No « S » to CUP SHAPED COTYLEDON.

- line 134: Fig4E doesn’t show that auxin peaks are located on the floral meristem.

- Fig2C: I have a problem with this figure. It seems that growth rates are lower at 20°C at the inter-sepal boundary than at 15°C, but the opposite is stated in the text line 173. Can the author clarify this by rephrasing the description of the experiment and the result. Remove « side views » in the legend of the Fig2C line 555.

- There is no citation of the Fig2G in the text.

- Concerning the expression of 5mCUC1 under the PTL promoter, can the author provide evidences that PTL is expressed in a more regular pattern? A description of the different expression patterns CUC and PTL would appreciated here has I’m not sure I get what more regular expression pattern means here.

---

## [Reviewer Report · Review: CUC/auxin patterning of decanalised petal number in *Cardamine hirsuta*
— R0/PR3]

The manuscript by Rambaud-Lavigne et al. entitled ´CUC/auxin patterning of decanalized petal number in Cardamine hirsute´, reports on petal number variation in C. hirsute. This is correlated with CUC1/2 expression, PIN1 and DR5. Furthermore, petal number with less variation was observed when plants were grown at 15 degrees Celcius.

The experiments are well-performed and results clearly presented. Imaging is beautiful. The data correlates with the conclusions.

Just a few minor comments for the authors:

L. 177 The use of the construct pChCUC2::3×GFP (ChCUC2), is a transcriptional GFP fusion, this would be good to add to the text.

Does temperature variation during flower development in Arabidopsis change petal number? And is there something known whether PIN1 and DR5 change in Arabidopsis upon different temperatures?

What would cause the different expression pattern of the CUC genes in C. hirsute compared to Arabidopsis? Any speculation?

---

## [Editor Report · Recommendation: CUC/auxin patterning of decanalised petal number in *Cardamine hirsuta*
— R0/PR4]

Dear Dr Hay and co-authors, 

We have now received the comments of the two reviewers of your manuscript. They agree - as I do - on the high quality and interest of the work. Some minor revisions are however required. Notably to clarify the number of replicates performed (actually, I could find the information in Methods, but I suggest to mention these numbers in the text or figures legends) and add statistical tests when possible as asked by Reviewer 1. Reviewer 2 accepted the manuscript, yet he has also minor comments, and his suggestions to enrich the discussion appear valuable. 

As a whole I recommend publication providing the minor revisions asked by the reviewers.

Thank you very much again for your nice contribution to Quantitative Plant Biology

Best regards

Daphné Autran

---

## [Reviewer Report · Review: CUC/auxin patterning of decanalised petal number in *Cardamine hirsuta*
— R1/PR8]

All my comments and queries have been taken into account in this new version of the manuscript therefore I’m happy with it. Although only the figure 2 has been modified, having the complete set of figures as well as the changes made to the main text in this revised version of the manuscript would have helped a lot the reviewing process.

---

## [Editor Report · Recommendation: CUC/auxin patterning of decanalised petal number in *Cardamine hirsuta*
— R1/PR9]

Dear Dr Hay and colleagues, 

We have now received the reviewers comments on the revised version of your manuscript. I agree with both of them that the comments were fully addressed and that the manuscript is ready for publication. 

Thank you again for your contribution to Quantitative Plant Biology.

Best regards